# Th2-dependent STAT6-regulated genes in intestinal epithelial cells mediate larval trapping during secondary *Heligmosomoides polygyrus bakeri* infection

**Stefanie Westermann[1], Christoph Schubart[1], Axel Dietschmann[2], Kirstin Castiglione[1], Daniel Radtke[1], David Voehringer**[1] *

**1** Department of Infection Biology, University Hospital Erlangen and Friedrich-Alexander Universität Erlangen-Nürnberg (FAU), Erlangen, Germany, **2** Junior Research Group Adaptive Pathogenicity Strategies, Leibniz Institute for Natural Product Research and Infection Biology—Hans Knöll Institute, Jena, Germany

* david.voehringer@uk-erlangen.de

**Data Availability Statement:** The datasets generated during and/or analysed during the current study are all included in the manuscript.

## Abstract

Gastrointestinal helminths are a major health threat worldwide. Alternatively activated macrophages (AAMs) have been shown to contribute to host protection during secondary helminth infections. AAMs express effector molecules that depend on activation of the IL-4- or IL-13-induced transcription factor signal transducer and activator of transcription 6 (STAT6). However, the specific role of STAT6-regulated genes like Arginase-1 (Arg1) from AAMs or STAT6-regulated genes in other cell types for host protection remains unclear. To address this point, we generated mice expressing STAT6 only in macrophages (Mac-STAT6 mouse). In the model of *Heligmosomoides polygyrus bakeri* (*Hpb*) infection, Mac-STAT6 mice could not trap larvae in the submucosa of the small intestine after secondary infection. Further, mice lacking Arg1 in hematopoietic and endothelial cells were still protected from secondary *Hpb* infection. On the other hand, specific deletion of IL-4/IL-13 in T cells blunted AAM polarization, activation of intestinal epithelial cells (IECs) and protective immunity. Deletion of IL-4Rα on IEC also caused loss of larval trapping while AAM polarization remained intact. These results show that Th2-dependent and STAT6-regulated genes in IECs are required and AAMs are not sufficient for protection against secondary *Hpb* infection by mechanisms that remain to be investigated.

## Author summary

Infection of mice with the gastrointestinal helminth *Heligmosomoides polygyrus bakeri* is widely used as a model of hookworm infection to study protective immunity. During secondary infection *Hpb* larvae are trapped in granulomas located in the submucosa of the small intestine by mechanisms that remain poorly defined. Alternatively activated macrophages (AAMs) and expression of STAT6-regulated genes, especially Arginase 1, were reported to play an important role in larval trapping. However, using various genetically

Raw data for bar graphs are provided in the supplement (Raw data graphs).

**Funding:** This work was supported by the Deutsche Forschungsgemeinschaft (DFG) grant CRC1181_A02 to D.V. The funder had no role in study design, data collection and analysis, decision to publish, or preparation of the manuscript.

**Competing interests:** All authors declare no conflict of interests.

modified mouse strains we show that AAMs are not sufficient and Arginase 1 is dispensable for larval trapping whereas T cell-derived IL-4/IL-13 and activation of STAT6-regulated genes in intestinal epithelial cells play a critical role in this process.

## Introduction

Infections with gastrointestinal helminths affect approximately 2 billion people worldwide and infected individuals, particularly children, often suffer from malnutrition accompanied by physical and cognitive deficits [1]. Efficient anthelminthic drugs for treatment are available, but rapid reinfection and increasing drug resistances underline the need for long-term preventive measures [1,2]. Unfortunately, the immune response against helminths is not yet fully elucidated but a complete understanding is required for development of effective new therapeutics and vaccines.

*Heligmosomoides polygyrus bakeri* (*Hpb*) is a rodent gut dwelling helminth that can be used to model human hookworm infection such as with *Ancylostoma ceylanicum* [3,4]. Infective L3 stage larvae are ingested and reach the small intestine within 24 to 48 hours, where they invade the submucosa until the outer muscularis layer to molt into L4 [3]. After 6 to 8 days post infection, L4 molt again to become adult worms and re-enter the intestinal lumen to mate whereupon females lay eggs that are released with the feces [3]. Notably, a primary infection with *Hpb* establishes chronicity in wild-type (WT) mice whereas after a secondary infection the larvae are trapped in a granulomatous structure in the submucosa, formed by a memory response of the host's immune system, largely preventing their re-entry as adults into the intestinal lumen [5].

Protective immunity against helminths is mediated by a type 2 immune response and involves an interplay of adaptive and innate immune cells. Interleukin (IL) IL-4 and IL-13 secretion from Th2 cells promotes STAT6-dependent polarization of alternatively activated macrophages (AAMs), which were shown to mediate protection against secondary infections with *Hpb* or *Nippostrongylus brasiliensis (Nb)* [6–8]. In addition, IL-4/IL-13 expressing innate immune cells, including eosinophils, basophils, mast cells and type 2 innate lymphoid cells (ILC2s), have also been reported to contribute to protection against gastrointestinal helminths by activation of tuft cells, goblet cells and smooth muscle cells in the small intestine [9–12].

AAMs are usually associated with wound repair, fibrosis and anti-helminth immune responses [5,13]. AAMs express several STAT6-induced effector molecules like Arginase 1 (Arg1), Resistin-like molecule alpha (Relm-α, *Retnla*), programmed cell death ligand 2 (PD-L2, *Pdcd1lg2*) and chitinase-like protein Ym-1 (*Chil3*) that have been suggested to contribute to host defense against helminths [7,8,14–16]. Arg1 has been shown to mediate worm expulsion through impairment of larval health and motility in a secondary *Hpb* infection [7]. On the other hand, *S. mansoni* infection in Arg1-deleted mice lead to enhanced type 2 immunity-mediated inflammation and increased death rate [17]. Relm-α was reported to have negative regulatory functions on type 2 immunity-mediated inflammation [18–20]. This effect could be linked to Relm-α expression in hematopoietic cells [21]. Furthermore, Ym-1 was shown to induce Relm-α expression in the lung and both molecules contribute to tissue repair [16]. PD-L2 expression by AAMs was shown to inhibit T cell proliferation and was suggested to prevent an overwhelming type 2 response [14]. Considering these diverse reports, the specific contribution of AAMs and their STAT6-regulated genes, particularly Arg1, to host protection in secondary helminth infections is not clear.

In this study, we generated Mac-STAT6 mice in which STAT6 can only be activated in macrophages (and some other LysM-expressing cells) to investigate the role of AAMs and their STAT6-regulated genes during secondary *Hpb* infection. STAT6 expression in macrophages was not sufficient and Arg1 expression was dispensable for larval trapping after secondary *Hpb* infection. In contrast, mice with constitutively active STAT6 in IECs were able to trap *Hpb* larvae, even in the absence of CD4[+] T cells. Importantly, selective deletion of IL-4/IL-13 in T cells or deletion of IL-4Rα on IEC both resulted in impaired larval trapping while AAM differentiation remained intact in the latter situation. These findings indicate that AAMs are not sufficient and Arg1 expression is dispensable for larval trapping while T cell-derived IL-4/IL-13 and activation of STAT6-regulated genes in IECs play a critical role.

## Results

### Generation of mice with STAT6 expression restricted to macrophages

To investigate the role of STAT6-regulated genes in macrophages for mediating immunity against helminths, we generated mice where STAT6 is expressed only in macrophages. We have previously shown that crossing CD19Cre or VillinCre mice to *Rosa26*[LSL-STAT6vt] mice leads to expression of a constitutively active form of STAT6 (STAT6vt) in B cells or intestinal epithelial cells (IEC), respectively [22–25]. Here, we crossed these *Rosa26*[LSL-STAT6vt] mice to LysMCre mice and subsequently to mice with a complete knockout of STAT6 (STAT6ko) in order to obtain mice where all cells are STAT6ko except those that express the LysM gene, i.e. predominantly macrophages and neutrophils (Mac-STAT6 mice). To test the phosphorylation of STAT6, we generated bone marrow derived macrophages (BMDMs) and isolated B cells from the spleen of Mac-STAT6, STAT6ko and WT mice and applied them to Western blot analysis. Contrary to expectations and for reasons that remain unclear, STAT6 in the BMDMs of Mac-STAT6 mice was not constitutively phosphorylated as can be seen in the Western blot of unstimulated cells (Fig 1A, left blot). However, when BMDM from the Mac-STAT6 mouse were stimulated with IL-4, STAT6 was phosphorylated comparably to STAT6 in BMDM from WT mice, meaning that STAT6 in the Mac-STAT6 mouse can be activated normally (Fig 1A, right blot). B cells isolated from the Mac-STAT6 mouse and stimulated with IL-4 did only show a very faint band, indicating a good specificity of STAT6 expression only in LysM-expressing cells. As expected, neither phosphorylated nor total STAT6 was detectable in BMDMs and B cells of STAT6ko mice. Furthermore, the expression of the STAT6-regulated genes *Arg1*, *Retnla* and *Pdcd1lg2* in BMDMs from Mac-STAT6 mice after IL-4 stimulation was comparable to that in WT mice and prominently enhanced compared to that in BMDMs from STAT6ko mice (Fig 1B).

In summary, the Mac-STAT6 mouse provides a tool to investigate the role of STAT6-regulated genes in macrophages without the contribution of STAT6-regulated genes in other cell types.

### AAMs are not sufficient for larval trapping during a secondary *Hpb* infection

In a secondary *Hpb* infection, polarization of AAMs requires memory Th2 cells [7,26]. To ensure sufficient induction of AAMs in the Mac-STAT6 mice, we decided to reinforce AAM polarization by administration of IL-4 complexes (IL-4c) as described elsewhere [27]. We performed a secondary *Hpb* infection with *i.p.* injection of IL-4c or PBS control every other day from the day of secondary helminth infection onwards (Fig 2A). IL-4c-treated Mac-STAT6 mice were not able to trap larvae during secondary infection (Fig 2B). In histological staining

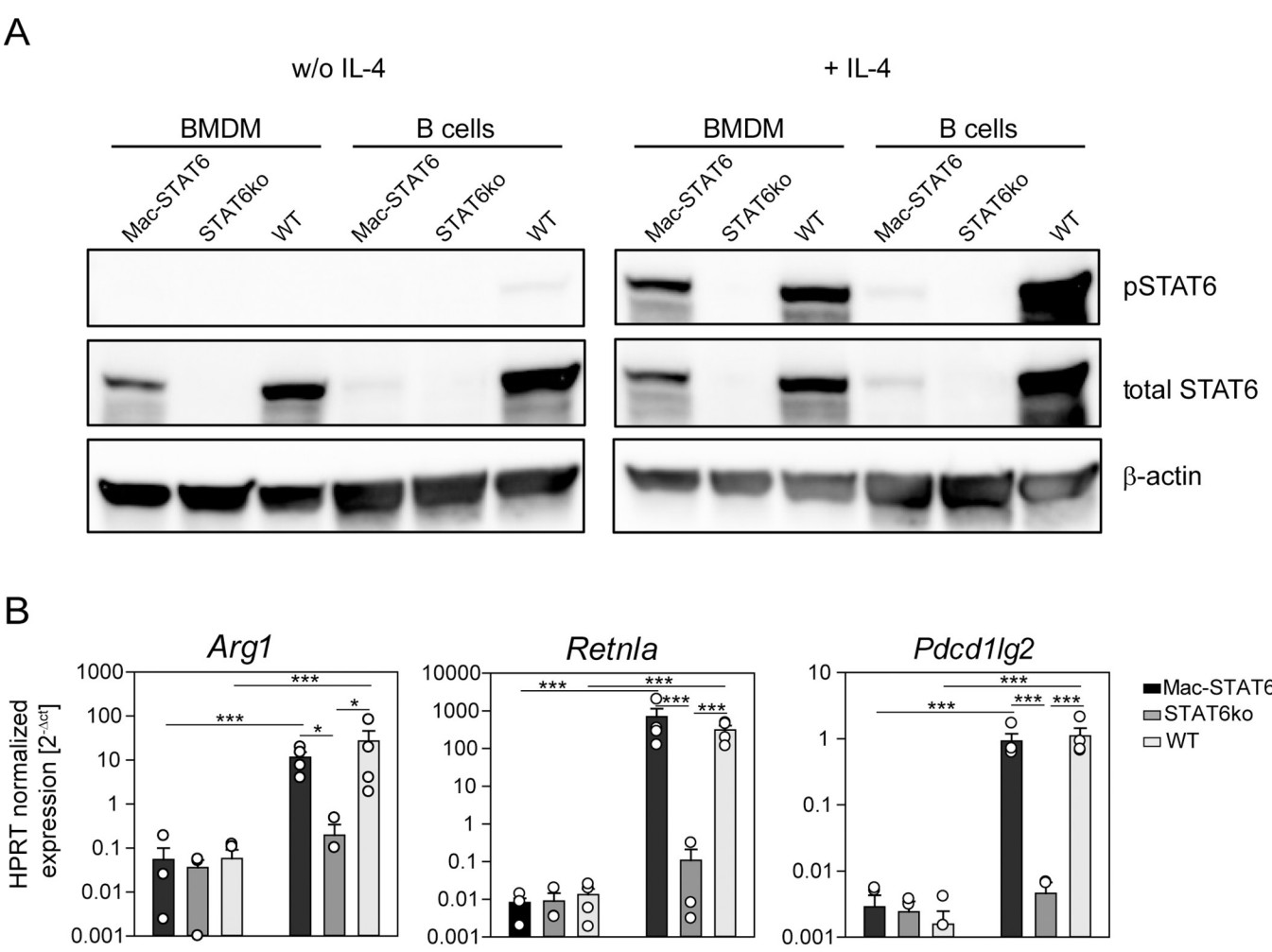

**Fig 1. Functional expression of STAT6 in macrophages of the Mac-STAT6 mouse.** Bone marrow derived macrophages (BMDM, purity >95%) were generated and splenic B cells (purity > 90%) isolated from mice that do express STAT6 only under control of the LysMCre promoter (Mac-STAT6), STAT6 knockout mice (STAT6ko) and BALB/c wild-type (WT) controls. A) Western blot of unstimulated (w/o) and IL-4-stimulated BMDM and B cell lysates of the indicated genotypes for phosphorylated STAT6 (pSTAT6), total STAT6 and β-actin as loading control. Western blot is representative for three to five mice per genotype from two independent experiments. B) qRT-PCR of BMDM from either Mac-STAT6, STAT6ko or WT mice stimulated with 20 ng/ml IL-4 for 48 hrs. Bars show mean expression + SEM of *Arg1*, *Retnla* and *Pdcd1lg2* normalized on *Hprt* of three to four mice per genotype out of two experiments. Statistical significance was determined by Two-Way-ANOVA with Holm-Sidak *post-hoc* testingon log-transformed data. ***$p < 0.001$; *$p < 0.05$.

of granuloma regions, Relm-α was abundantly expressed in IL-4c-treated Mac-STAT6 mice, more prominent than in the PBS or STAT6ko controls and co-staining with CD68 and DAPI confirmed Relm-α expression in AAMs with similar frequency in Mac-STAT6 and WT mice (Fig 2C). In contrast, compared to WT mice fewer macrophages expressed Arg1 in Mac-STAT6 and STAT6ko mice in all different conditions (Fig 2D). Flow cytometry analysis of cells from the peritoneal cavity (PEC) was used to determine the efficiency of AAM polarization of PEC macrophages, defined by increased PD-L2 surface expression (S1A and S1B Fig). PD-L2 was upregulated significantly in IL-4c-treated Mac-STAT6 mice compared to STAT6ko mice and upregulation was comparable to the level in WT mice (S1C Fig). In addition, RT-PCR of sorted PEC macrophages from IL-4c-treated Mac-STAT6 mice revealed their expression of other AAM-associated markers (S1D Fig). We further analysed mice on day 2

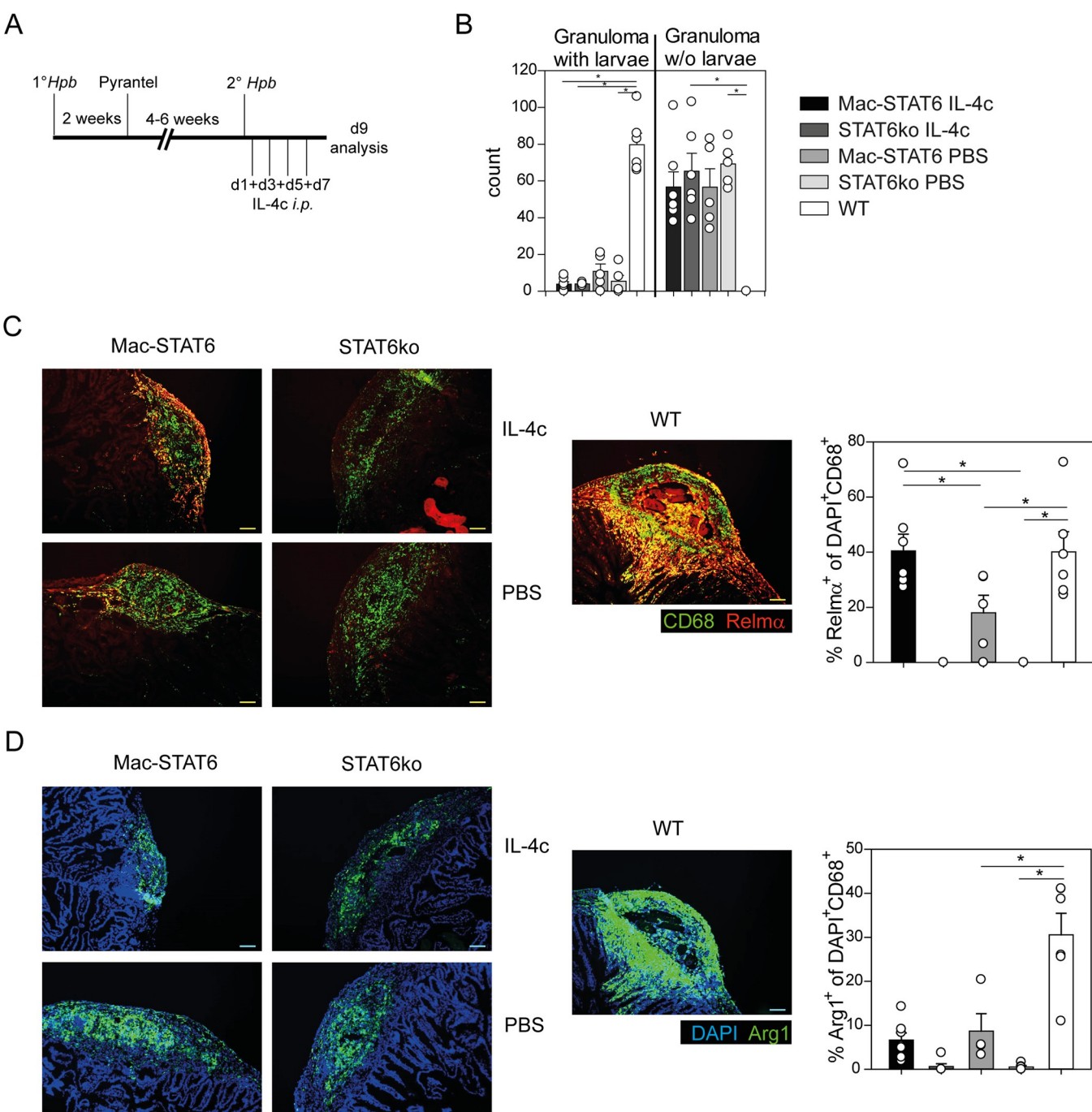

**Fig 2. AAMs are not sufficient for larval trapping during a secondary *Hpb* infection.** A) Schematic timeline: Mac-STAT6 mice, STAT6ko mice or BALB/c (WT) control mice were primary infected with *Hpb*, infection was cleared by Pyrantel pamoate after two weeks and four to six weeks later, mice were secondarily infected. 5 μg/mouse IL-4c in 100 μl PBS or PBS for control groups was injected *i.p.* on days 1, 3, 5 and 7 after secondary infection. B) Mean + SEM of granuloma with or without (w/o) larvae on day 9 after secondary infection. Statistical analysis was performed separately for granuloma with or w/o larvae by Kruskal-Wallis with Dunn's *post-hoc* testing. *p<0.05. Data are pooled from three independent experiments with five to seven mice per group. C and D) Histological stainings and quantifications of small intestinal cross-sections day 9 after secondary *Hpb* infection for detection of Relm-α (red) and CD68 (green) (C), or DAPI (blue) and Arg1 (green) (D). Quantification plots display Mean + SEM of percentage of Relm-α⁺ or Arg1⁺ of DAPI⁺CD68⁺ macrophages, respectively. Representative picture and quantification for four to seven mice per genotype. Scale bar is 100 μm.

after secondary *Nb* infection. Compared to STAT6ko mice, both Mac-STAT6 and WT mice showed a trend to fewer larvae in the lung and expressed similar levels of AAM markers (S2 Fig). This indicates that AAM polarization can be induced in macrophages of Mac-STAT6 mice in another helminth infection model. It has been described before that *Hpb*-specific antibodies of the IgG2a/c isotype can induce Arg1 expression in macrophages and inhibit the motility of larvae [28]. Therefore, we determined the humoral immune response in Mac-STAT6 and control mice. Due to the deficiency of STAT6 in B cells of Mac-STAT6 mice, we observed lower serum levels of total and antigen-specific IgG1 and IgE but higher levels of IgG2a and IgG2b (S3 and S4 Figs). This shows that humoral immunity is not impaired but altered in Mac-STAT6 and STAT6ko mice which actually produce more of the reported macrophage activating isotype. However, we cannot exclude that the quality of the antibodies (affinity, glycosylation pattern) is different compared to WT mice.

Together, these data suggest that AAMs are not sufficient for larval trapping in a secondary *Hpb* infection. However, the lower expression level of Arg1 in Mac-STAT6 mice may indicate that Arg1 plays an important role.

## Arg1 is dispensable for larval trapping

Arg1 was reported to be essential for larval trapping, as mice that were given the Arginase inhibitor BEC were not able to trap the larvae in a secondary *Hpb* infection [7]. However, a systemically applied inhibitor can have many undesired side effects, so we decided to verify that previous observation with a genetic deletion mouse model. The Tie2Cre_Arg1$^{fl/fl}$ mice have a genetic knockout of Arg1 in all hematopoietic and endothelial cells and were described before [29]. BMDMs of Tie2Cre_Arg1$^{fl/fl}$ mice did confirm the knockout by the absence of Arg1 induction upon IL-4 stimulation (Fig 3A). Other IL-4-inducible genes such as *Retnla*, *Chil3*, *Pdcd1lg2*, *Mrc1* and *Mmp12* were regulated similarly in the IL-4 stimulated BMDM of Tie2Cre_Arg1$^{fl/fl}$ and Arg1$^{fl/fl}$ control mice. To investigate the role of Arg1 *in vivo*, we performed a secondary *Hpb* infection. Unexpectedly, the Tie2Cre_Arg1$^{fl/fl}$ mice were able to trap larvae as efficient as their Arg1$^{fl/fl}$ littermate controls (Fig 3B). Histological staining for Arg1 and DAPI did show the trapped larvae in the submucosa although Arg1 is absent in the Tie2-Cre_Arg1$^{fl/fl}$ mouse (Fig 3C, right panel). Besides, AAMs, defined here by Relm-α and CD68 co-expression, were present in the granuloma in a similar amount in mice of both genotypes (Fig 3C, upper panel).

Taken together, genetic deletion of Arg1 in hematopoietic and endothelial cells did not result in impaired larval trapping.

## Activated STAT6 in intestinal epithelial cells is sufficient for larval trapping in the absence of CD4$^+$ T cells

Since STAT6-regulated genes in AAMs appeared insufficient for larval trapping in a secondary *Hpb* infection, we investigated the role of STAT6 in other cell types. A previous report from our group introduced the VillinCre_STAT6vt mice, which express a constitutively active form of STAT6 in intestinal epithelial cells (IECs) which causes accumulation of secretory IECs [22]. Remarkably, these mice show a protective immune response against *Hpb* already during primary infection [22]. One may hypothesize that activated IEC cause larval trapping by promoting a local Th2 response. Considering that memory CD4$^+$ T cells were reported to be essential for larval trapping in secondary *Hpb* infection [7], we asked whether VillinCre_STAT6vt would still be protected from secondary infection if CD4$^+$ T cells are absent. Thus, we performed a secondary *Hpb* infection and depleted CD4$^+$ T cells by *i.v.* injection of an anti-CD4 antibody on the same day as helminth infection (day 0) and again on day 4 after infection

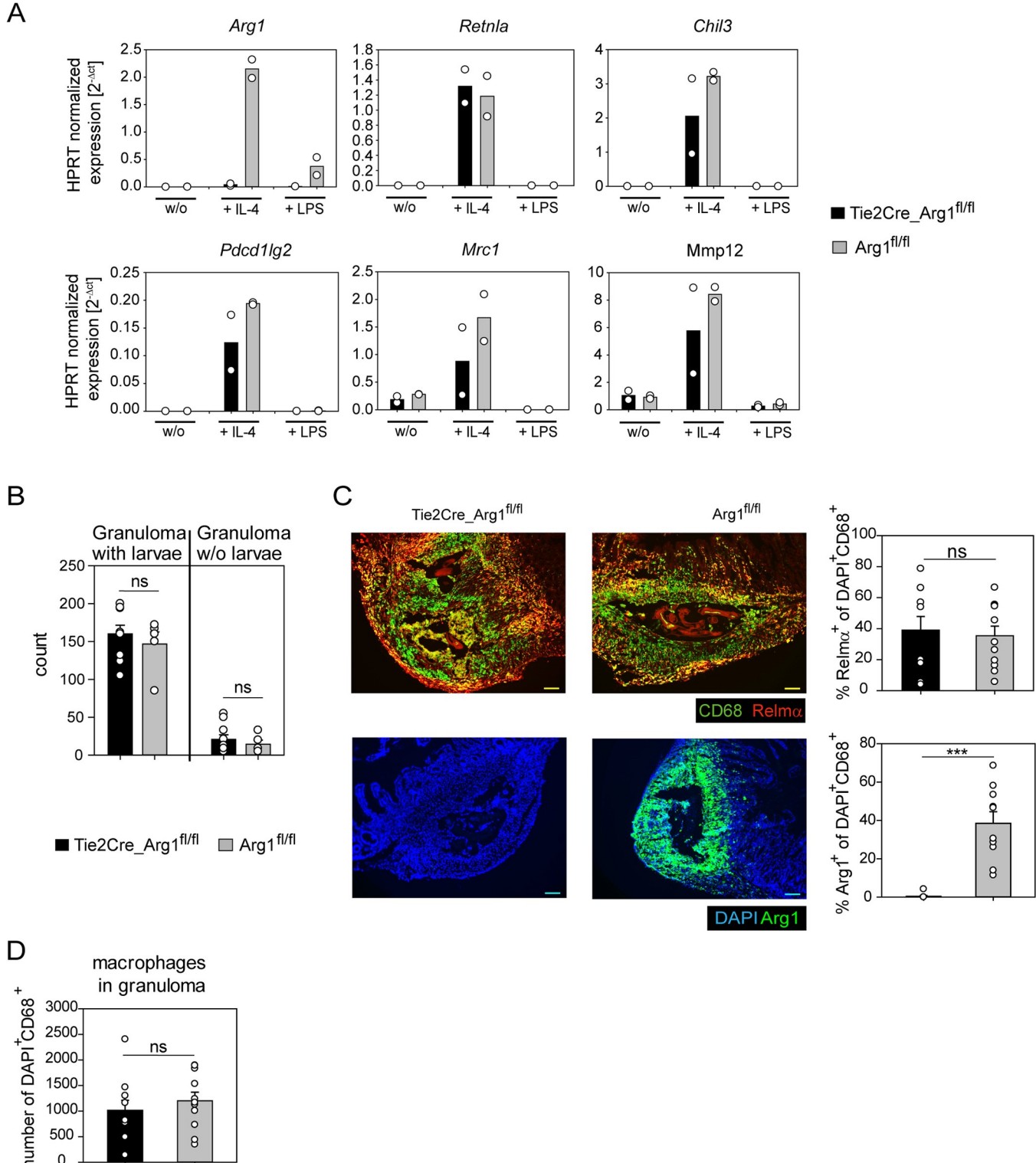

**Fig 3. Arg1 is dispensable for larval trapping.** A) qRT-PCR of BMDM from mice that lack Arg1 (Tie2Cre_Arg1^fl/fl) or control mice (Arg1^fl/fl) stimulated with 20 ng/mL IL-4 or 100 ng/mL LPS for 48 hrs. Bars show mean for two mice per genotype and condition from one experiment. B) Granuloma count on day 9 after secondary *Hpb* infection in the small intestinal submucosa of Tie2Cre_Arg1^fl/fl and Arg1^fl/fl mice. Mean + SEM number of granuloma with larvae and granuloma without (w/o) larvae of five to ten mice per group pooled from two independent experiments. C) Immunofluorescence staining from small intestinal tissue day 9 after secondary *Hpb* infection. The upper panel shows staining for Relm-α (red) and CD68 (green) and the lower panel staining for Arg1 (green) and DAPI (blue). Scale bar is 100 μm. Pictures are representative for ten to eleven mice per genotype from two independent experiments.

Quantifications display Mean + SEM of percentage of Relm-α⁺ or Arg1⁺ of DAPI⁺CD68⁺ cells, respectively. D) Quantification of total macrophages (DAPI⁺CD68⁺) in granuloma. Statistical significance between the different genotypes was determined by Student's t-Test, ***p < 0.001; ns not significant.

(Fig 4A). Depletion of CD4⁺ T cells occured with high efficiency while macrophages and B cells remained largely unaffected (S5 Fig). Strikingly, the VillinCre_STAT6vt mice were still able to trap larvae, significantly better than their STAT6vt littermate controls, despite the absence of CD4⁺ T cells (Fig 4B). Accordingly, CD4⁺ T cell-depleted VillinCre_STAT6vt mice had significantly lower fecal egg counts than CD4⁺ T cell-depleted STAT6vt mice on day 14 after secondary infection (Fig 4C). Interestingly, gene expression of *Arg1*, *Retnla* and *Chil3* were strongly reduced in CD4⁺ T cell-depleted compared to isotype-treated VillinCre_S-TAT6vt mice, indicating that the typical AAM-associated gene expression was diminished (Fig 4D). Also, histological stainings for Relm-α and Arg1 confirmed the very low expression of AAM effector proteins in the absence of CD4⁺ T cells (Fig 4E). In contrast, the expression of *Retnlb*, a STAT6-regulated gene expressed in goblet cells (a subset of IECs), was only reduced in CD4⁺ T cell-depleted STAT6vt control but not VillinCre_STAT6vt or isotype control-treated mice (Fig 4D).

In summary, we confirm here that CD4⁺ T cells are essential for larval trapping and activation of IECs in secondary *Hpb* infection. However, the need for CD4⁺ T cells could be overcome by expression of a constitutively active form of STAT6 in IECs, emphasizing the prominent role of STAT6-regulated genes in IECs in contributing to larval trapping.

## STAT6 signaling in IECs is required for efficient larval trapping

Since our data from the VillinCre_STAT6vt mice suggested that STAT6 signaling in IECs is important for larval trapping during secondary *Hpb* infection, we asked next whether larval trapping would be impaired when STAT6 signaling in IECs is interrupted. Therefore, we generated mice in which the IL-4Rα is selectively deleted in IECs (VillinCre_IL-4Rα^fl/fl). Strikingly, the VillinCre_IL-4Rα^fl/fl mice trapped the larvae significantly less efficient than their Cre⁻ littermate controls upon secondary infection (Fig 5A). Histology sections of granuloma regions in the small intestine revealed that Relm-α- and Arg1-expressing macrophages were well present around the granuloma with similar numbers in VillinCre_IL-4Rα^fl/fl and IL-4Rα^fl/fl control mice (Fig 5B). Furthermore, there were no significant differences in the compartments of eosinophils, macrophages, B cells, CD4⁺ and CD8⁺ T cell in the PEC and mLN of VillinCre_IL-4Rα^fl/fl compared to controls (S6 Fig). However, activation of IECs was abrogated in VillinCre_IL-4Rα^fl/fl mice as indicated by a complete loss of Relm-β⁺ IECs (Fig 5B).

In conclusion, STAT6-regulated genes in IECs are required for efficient larval trapping but dispensable for AAM polarization in granuloma.

## IL-4/IL-13 from CD4⁺ T cells is required for activation of IECs, AAM polarization and larval trapping

IL-4 and IL-13 are the two major cytokines for protective immunity against gastrointestinal helminths. To investigate whether T cell-derived IL-4/IL-13 is required for IEC activation *in vivo*, we performed a secondary *Hpb* infection with mice that have a T cell-specific knockout of IL-4/IL-13 (4-13Tko), complete IL-4/IL-13 knockout (4-13ko) and WT control mice. Neither 4-13Tko nor 4-13ko mice were able to trap larvae in the submucosa (Fig 6A). Gene expression analysis of whole small intestinal tissue showed that expression of *Retnlb* was not induced in both knockout mice while it was highly expressed in WT mice (Fig 6B), indicating that IECs are not activated in the knockout mice. Similarly, the AAM-associated markers

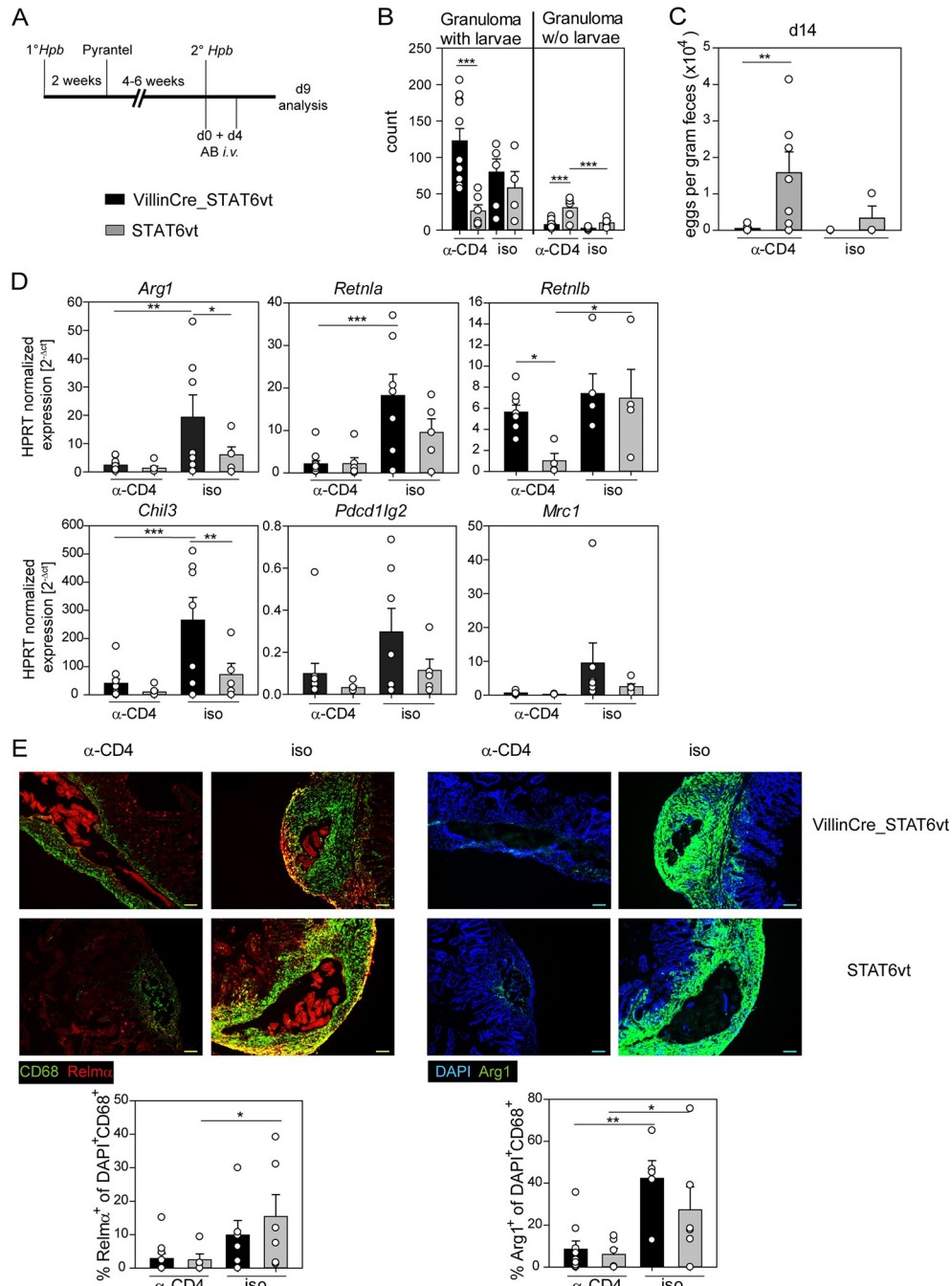

**Fig 4. Activated STAT6 in intestinal epithelial cells is sufficient for larval trapping in the absence of CD4[+] T cells.**
A) Schematic timeline for the CD4 depletion experiment performed with mice with constitutively active STAT6 in IECs (VillinCre_STAT6vt) and Cre[-] littermate controls (STAT6vt): Primary *Hpb* infection was cleared by oral gavage with twice 1 mg Pyrantel pamoate per mouse. 4–6 weeks later, mice were secondarily infected with *Hpb* and anti-CD4 or isotype antibody (AB) was injected *i.v.* on day 0 and 4 of secondary infection. B) Number of counted granuloma with or without (w/o) larvae in the submucosa on day 9 or 10 after secondary *Hpb* infection. Bars show mean count + SEM of four to eleven mice per group from two independent experiments. C) Eggs per gram feces counted on day 14 after secondary *Hpb* infection. Data show mean count + SEM of three to seven mice per group from two independent experiments. D) Relative expression of *Arg1*, *Retnla*, *Retnlb*, *Chil3*, *Pdcd1lg2*, and *Mrc1*, calculated by normalisation to *Hprt*. Bars show mean + SEM of four to eleven mice per group from two independent experiments. E) Histological staining and quantification of Relm-α (red) and CD68 (green) in tissue sections from small intestine day 9 after secondary *Hpb* infection in the left panel and for DAPI (blue) and Arg1 (green) in the right panel. Mean + SEM of percentage of Relm-α[+] or Arg1[+] of DAPI[+]CD68[+] cells, respectively. Representative picture for one mouse per

genotype and quantification for all five to ten mice per group from two independent experiments. Scale bar corresponds to 100 μm. B-E) Statistical significance was determined by Two-Way ANOVA with Holm-Sidak *post-hoc* testing. ***p < 0.001; **p < 0.01; *p < 0.05.

*Arg1*, *Retnla*, *Chil3*, *Pdcd1lg2* and *Mrc1* and the Relm-α-induced enzyme Lysyl-hydroxylase 2 (*Lh2*) in fibroblasts involved in formation of collagen bundles were only expressed in WT but not in 4-13Tko or 4-13ko mice (Fig 6B). However, expression of phospholipase A2 group 1b (*Pla2g1b*) in IECs, which was previously shown to contribute to larval killing during secondary *Hpb* infection [30], was comparable in all three groups of mice. Histological staining for Relm-α (Fig 6C, upper panel) and Arg1 (Fig 6C, lower panel) confirmed the gene expression data also on protein level. We further observed impaired differentiation of goblet cells (Relm-β⁺) and tuft cells (DCLK1⁺) while Paneth cells (Lysozyme⁺) were not affected and the number of IECs in cell cycle (Ki67⁺) were even increased (S7 Fig).

Taken together, IL-4/IL-13 from CD4⁺ T cells are required for activation of IECs, differentiation of AAMs and essential for larval trapping in a secondary *Hpb* infection.

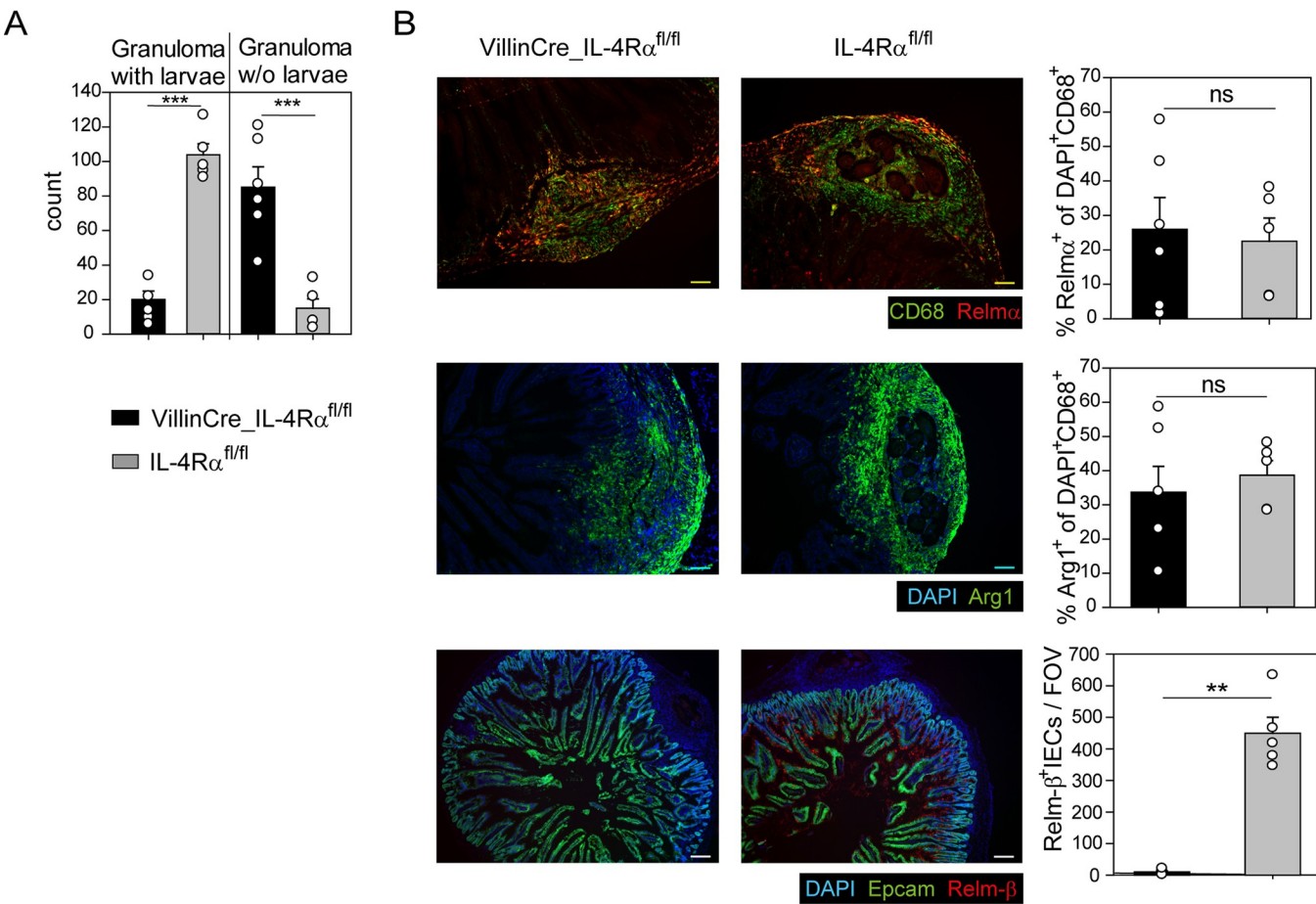

**Fig 5. STAT6 signaling in IECs is required for efficient larval trapping.** Mice with a specific deletion of the IL-4Rα in intestinal epithelial cells (VillinCre_IL-4Rα^fl/fl) and their Cre⁻ littermate controls (IL-4Rα^fl/fl) were subjected to a secondary *Hpb* infection. A) Number of granuloma in the submucosa with or without (w/o) larvae counted on day 9 after secondary *Hpb* infection. B) Representative pictures and quantification of Relm-α (upper panel), Arg1 (middle panel) and Relm-β (lower panel) in histological small intestinal sections on day 9 after secondary *Hpb* infection. Quantification was calculated in Fiji (Relm-α and Arg1) or manually by counting Relm-β⁺ IECs per field of view (FOV). Scale bar corresponds to 100 μm (Relm-α and Arg1) or 200 μm (Relm-β). A-B) Mean + SEM of five to six mice per group from two independent experiments are displayed. Statistical significance was determined by Student's t-Test. ***p < 0.001; **p < 0.01; ns not significant.

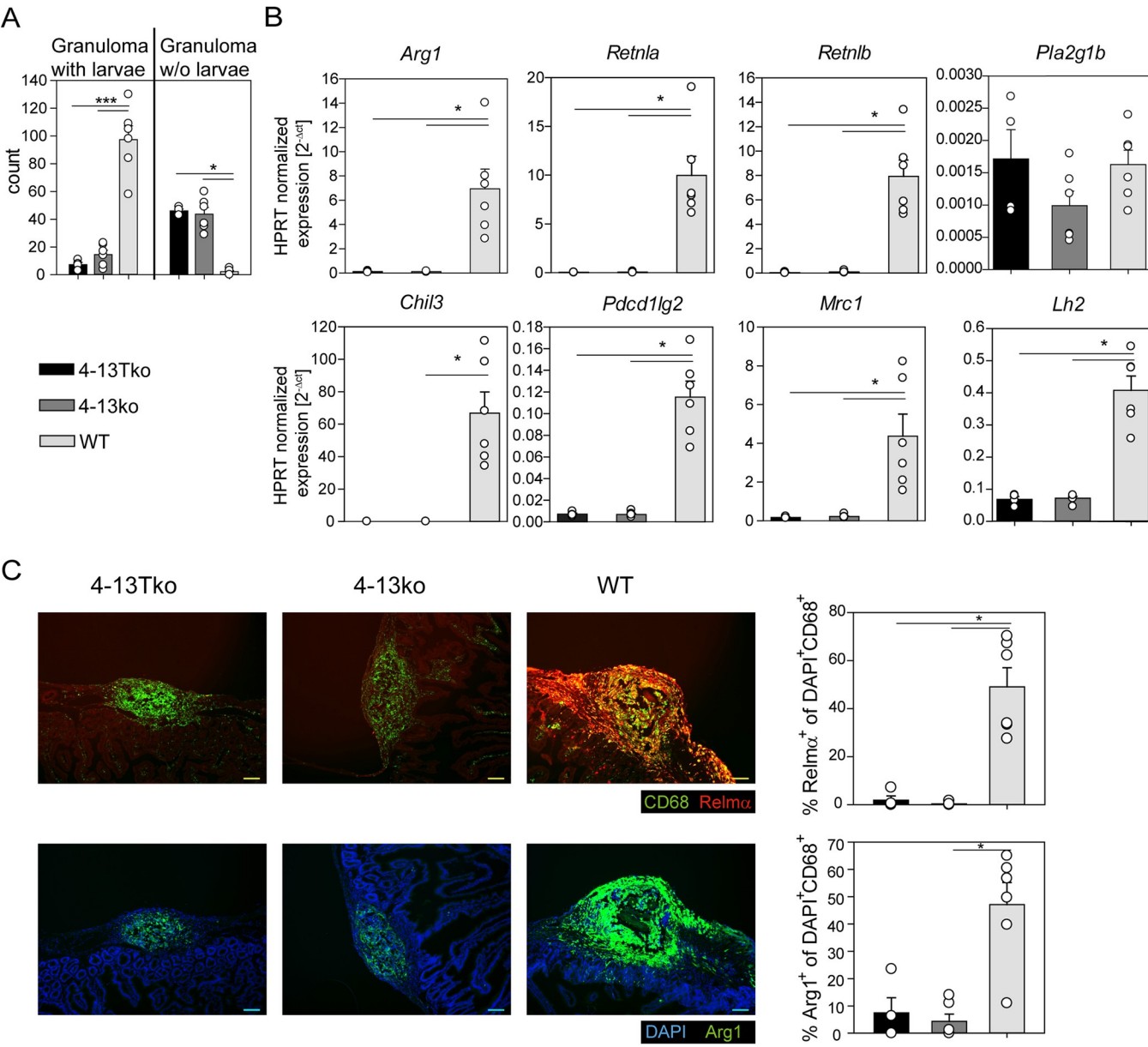

**Fig 6. IL-4/IL-13 from CD4$^+$ T cells is required for activation of IECs, AAM polarization and larval trapping.** T cell-specific IL-4/IL-13 knockout (4-13Tko), complete IL-4/IL-13 knockout (4-13ko) or B6 wild-type control (WT) mice were infected twice with *Hpb* and analysed on day 9 after secondary infection. A) Number of counted granuloma with or without (w/o) larvae. B) qRT-PCR data from whole small intestinal tissue for *Arg1*, *Retnla*, *Retnlb*, *Pla2g1b*, *Chil3*, *Pdcd1lg2*, *Mrc1, and Lh2*. C) Immunofluorescence staining in small intestinal sections day 9 after secondary *Hpb* infection. Upper panel was stained for Relm-α (red) and CD68 (green) and lower panel for DAPI (blue) and Arginase 1 (green), scale bar is 100μm. Bars display quantification of percentage of Relm-α$^+$ or Arg1$^+$ cells of DAPI$^+$CD68$^+$. Representative pictures and quantification for four to six mice per genotype from two independent experiments. A-C) Bars show mean + SEM of data pooled from two independent experiments with four to six mice per group. Statistical significance was determined by One-Way ANOVA with Holm-Sidak *post-hoc* testing or, if normality or equal variance were not given, by Kruskal-Wallis with Dunn's *post-hoc* testing. ***p $< 0.001$; *p $< 0.05$.

## Discussion

AAM polarization and expression of several effector molecules is dependent on macrophage-intrinsic activation of STAT6. We generated a mouse model where STAT6-regulated genes could only be activated in macrophages and observed that AAMs and their STAT6-regulated

genes were dispensable for larval trapping during secondary *Hpb* infection. Instead, T cell-derived IL-4/IL-13 and activation of STAT6 in IECs were critical components for larval trapping.

AAMs are induced by IL-4-secreting memory Th2 cells in the context of helminth infections and were described to mediate protection not only during secondary *Hpb* infection but also during secondary infection with *Nippostrongylus brasiliensis* (*Nb*) [7,8,31,32]. It has been reported that during secondary *Hpb* infection of WT mice, larvae are trapped in the small intestinal submucosa through memory Th2 cell-polarized AAMs and their expression of Arg1 [7]. In this study polarization of AAMs required cell-intrinsic expression of STAT6, as CD4[+] T cell transfer from primary *Hpb* infected WT mice into STAT6ko mice did not sufficiently induce AAMs and mice retained higher parasite and egg burden than WT recipients [7]. Administration of IL-4c has been shown to induce Relm-α expression in lung interstitial and PEC macrophages [27]. We administered IL-4c during secondary *Hpb* infection to successfully induce AAM polarization, as shown by flow cytometry, RT-PCR and histology. However, we did not observe that these AAMs were sufficient to trap the larvae. A previous study has shown that IL-4c administration to WT mice during the first few days of a primary *Hpb* infection is also not sufficient for larval trapping [33]. The reported requirement of Arg1 expression from AAMs in protection during secondary *Hpb* infection was concluded from experiments using an arginase inhibitor [7]. Systemic administration of such chemical inhibitors can have side effects that influence the outcome of the infection. This issue can be avoided by genetically deleting Arg1. In that case, we found no evidence for a protective role of Arg1 in secondary *Hpb* infection.

Although our mouse model with selective STAT6 expression in macrophages showed that this is not sufficient to trap larvae, it implicates that STAT6 expression in other cell types must be important. Others have already shown that IL-4 receptor expression on non-bone marrow-derived cells is required for worm expulsion from the intestine [34]. Further, hypercontractility of smooth muscle cells evoked by IL-4 and IL-13 was shown to be dependent on STAT6 in helminth infection [35]. Previous data from our group did show that selective expression of constitutively active STAT6 in IECs is protective against *Nb* and *Hpb* infections and leads to trapping of the larvae already in primary *Hpb* infection [22]. As expected, we also observed trapping of larvae in these mice during secondary *Hpb* infection, similar to WT mice. Remarkably, when CD4[+] T cells were depleted during secondary *Hpb* infection, mice with the constitutively active STAT6 in IECs were still protected, whereas WT mice were not. These data confirm the critical role of CD4[+] T cells for larval trapping after secondary infection as shown before [7,26]. Notably, expression of AAM-associated markers like Arg1, Relm-α or Ym-1 was not induced in the CD4[+] T cell-depleted VillinCre_STAT6vt mice, suggesting that STAT6-regulated genes in IECs are sufficient to protect mice in the absence of Th2 cells and AAMs. In turn, these data also show that STAT6-regulated genes in other cells are redundant if STAT6 in IECs is activated. Importantly, VillinCre_IL-4Rα[fl/fl] mice could not trap larvae during secondary infection despite presence of AAMs in granuloma. Others have already shown the importance of IL-4 receptor signaling in IECs as VillinCre-IL4Rα[flox/-] mice exhibited impaired worm expulsion in primary *Nb* infection [36]. Further, a recent study has reported that STAT6 phosphorylation and O-GlcNAcylation leads to upregulation of the STAT6-regulated gene Gasdermin C in IECs after primary *Hpb* infection, which facilitated IL-33 secretion via membrane pores and thereby contributed to effective anti-helminth response and intestinal inflammation [37]. Whether this mechanism is also important during a secondary *Hpb* infection remains to be examined. Relm-β is an effector molecule secreted by goblet cells and has been reported to protect mice from *Hpb* infection through inhibiting fecundity and survival of adult worms [36,38]. We observed high *Retnlb* expression in the VillinCre_STAT6vt mice, even

after depletion of CD4$^+$ T cells. However, it is rather unlikely that Relm-β contributes to larval trapping because it is secreted to the intestinal lumen while L4 stage larvae are trapped in granuloma structures in the submucosal layer. In our study, Relm-β rather serves as a marker for IL-4/IL-13-induced activation of IECs and goblet cell hyperplasia. Goblet cells and tuft cells were strongly reduced in 4-13Tko mice after secondary *Hpb* infection demonstrating the requirement of Th2 cells for IEC activation in this model. This is in contrast to the *Nb* infection model where ILC2- rather than T cell-derived IL-4/IL-13 is required for IEC activation and worm expulsion [39].

In conclusion, we demonstrate the importance of an AAM-independent Th2-IEC-axis for protection against secondary *Hpb* infection and the relevant IEC-derived molecules that orchestrate larval trapping remain to be identified.

## Materials and methods

### Ethics statement

Mouse experiments were conducted with permission of authorities from the Government of Lower Franconia, Germany.

### Mice

Generation of STAT6vt mice was described before [22]. These mice were then mated with LysMCre mice to have the STAT6vt expressed under the control of the promotor of lysozyme M [40]. LysMCre-STAT6vt mice were then crossed to mice with a complete knockout of STAT6 (4get STAT6ko) [41] to receive a mouse line where STAT6vt can only be expressed in LysM expressing cells (mainly macrophages and neutrophils) and with a STAT6ko phenotype for all LysM non-expressing cell types, termed Mac-STAT6 mouse. Mac-STAT6 and STAT6ko mice were on a BALB/c background and BALB/c WT mice (Charles River, Sulzfeld, Germany) were used as controls. Mice with a knockout of Arg1 in hematopoietic and endothelial cells, Tie2Cre-Arg1$^{fl/fl}$ mice, were described before and on BALB/c background [29]. VillinCre_STAT6vt mice on C57BL/6 background, which express a constitutively active form of STAT6 in intestinal epithelial cells, were described before [22]. VillinCre_IL-4Rα$^{fl/fl}$ mice on C57BL/6 background were generated by crossing Cre-negative littermates from the VillinCre_STAT6vt line to IL-4Rα$^{fl/fl}$ mice [42]. Mice with a T cell-specific knockout of IL-4/IL-13 (CD4Cre_IL-4/IL-13$^{fl/fl}$, named here 4-13Tko) [43] and mice with a complete knockout of IL-4/IL-13 (4-13ko) [44] were described before. Both lines were on B6 background and C57BL/6J or Ly5.1 B6 mice (Charles River) were used as control. All mice were kept under specific pathogen free conditions.

### *Hpb* infection

Mice were given 200 L3 stage larvae by oral gavage. For a secondary *Hpb* infection, chronic primary infection was cleared by administration of 1 mg pyrantel pamoate (Sigma-Aldrich) per mouse on day 14 and 16 after primary infection. After 4–6 weeks, mice were challenged with 200 L3 stage larvae given by oral gavage. To assess parasite egg burden, fecal pellets were collected and soaked in 1 mL water overnight. After addition of 3 mL saturated sodium chloride (Merck Millipore, Darmstadt, Germany), eggs were counted in a McMaster counting chamber (FiBL, Frick, Switzerland) and normalized to gram of feces. *Hpb* excretory/secretory antigen (HES) was prepared from adult worms isolated from the small intestine of infected mice. Isolated worms were washed extensively with PBS/1% Pen-Strep (v/v) and 1000 worms/mL were incubated in RPMI/1% Pen-Strep (v/v) at 37°C and 5% $CO_2$ for five days. After incubation, supernatant was taken, sterile filtered and stored at -80°C.

## *Nb* infection

Mice were infected with *Nippostrongylus brasiliensis* (*Nb*) by subcutaneous injection of 500 L3 stage larvae at the base of the tail and treated with antibiotics (2 g/L neomycin sulfate, 100 mg/L polymyxin B) in the drinking water for the first 5 days after infection. Secondary infection was performed more than 28 days after primary infection and mice were sacrificed on day 2 after secondary infection. The number of larvae that reached the lung was determined by allowing the larvae to migrate out of the tissue and subsequent counting of the larvae.

## IL-4 complex injection

Preparation and usage of IL-4 complex (IL-4c: anti-IL-4 antibody to IL-4) was performed as described before [27,45]. Briefly, recombinant mouse IL-4 (Biolegend, San Diego, CA) was complexed to anti-IL-4 mAB (11B11, BioXcell, West Lebanon, NH) at a 1:5 ratio of molecular weight and 5 μg IL-4 complexed to anti-IL-4 in 100 μL PBS or PBS only as control were injected *i.p.* on day 1, 3, 5 and 7 after secondary *Hpb* infection.

## CD4$^+$ T cell depletion

On day 0 and day 4 of secondary *Hpb* infection, 250 μg of anti-CD4 (GK1.5, BioXcell) or iso-type control (Rat IgG2b, LTF-2; BioXcell) were injected *i.v.* into the tail vein of the mice. Depletion of CD4$^+$ T cells was verified on day 8 by flow cytometry of blood samples.

## Bone marrow derived macrophages (BMDM)

For BMDM generation, femurs and tibiae from mice were flushed out and cells were subjected to hypotonic red blood cell lysis. $3 \times 10^6$ cells were seeded on a 10 cm petri dish and cultured in DMEM (10% FCS, 1% L-Glutamine, 1% Pen-Strep, all (v/v)) containing 15% (v/v) L929 supernatant with medium changed on day 3 and 5. Differentiated macrophages were harvested on culture day 7 and $1 \times 10^6$ cells per mL were seeded in a 24-well plate and stimulated with 20 ng/mL IL-4 or 100 ng/mL LPS or left unstimulated. For Western blot, stimulation was performed for 1 hr, whereas for qRT-PCR stimulation was performed for 48 hrs, as indicated in the figure legends. Purity of BMDM cultures was determined by flow cytometry and only cultures with >95% CD11b$^+$ F4/80$^+$ cells were used for further experiments.

## B cell isolation

Spleens from at least two mice per genotype were pooled and meshed through a 70 μm cell strainer. Single cell suspension was resuspended in Mojo Sort Buffer (2.5% BSA (w/v), 10 mM EDTA, PBS) and B cells were isolated using magnetic bead isolation with the MojoSort Pan B cell Isolation Kit (BioLegend, San Diego, CA) and an EasySep Magnet (Stemcell Technologies, Vancouver, Canada) according to the manufacturer's protocol. $10 \times 10^6$ B cells per condition were stimulated in DMEM (10% FCS, 1% L-Glutamine, 1% Pen-Strep, all (v/v)) with 20 ng/ml IL-4 for 1 hr for Western blot analysis. Purity of B cell isolation was determined by flow cytometry and was >90%.

## Sort of PEC macrophages

Mac-STAT6, STAT6ko and WT mice were treated with IL-4c *i.p.* on day 0 and day 2 and sacrificed on day 4, as described before [27]. PEC cells were harvested in PBS and stained with F4/80-PE-Cy7 (BM8) and CD11b-FITC (M1/70) as described in the flow cytometry section. Macrophages were sorted as F4/80$^+$CD11b$^+$ on a S3 sorter (Bio-RAD, Hercules, CA). Cells were sorted into RPMI/40% FCS (v/v), pelleted, lysed in RLT/10% DTT (v/v) and stored at –80˚C

until RNA preparation for qRT-PCR analysis. Purity was determined after sort and only samples containing >90% F4/80⁺CD11b⁺ cells were used for gene expression analysis.

## ELISA

For detection of antibodies in serum, plates were coated with rat-anti-mouse IgE (R35-72; BD) or goat-anti-mouse IgG (Southern Biotech) overnight at 4˚C. Purified mouse IgE (BD) or unlabelled mouse IgG1, IgG2a or IgG2b (all Southern Biotech) were used to generate a standard curve. Goat-anti-mouse IgE-AP, IgG1-AP, IgG2a-AP or IgG2b-AP (all Southern Biotech) were used for detection. For detection of parasite-specific antibodies, plates were coated with 20 µg/mL of HES suspension overnight at 4˚C. After blocking for 2 hrs with 3% BSA, samples were incubated at room temperature for 2 hrs and the AP-coupled antibodies named above were used for detection. Absorption was measured on a Multiscan FC photometer (Thermo Fisher) at 405 nm and blank wells were used to subtract background.

## Western blot

Cells were washed with PBS and then lysed in RIPA lysis buffer (1% (v/v) NP-40, 50 mM Tris (pH7.4), 0.15 M NaCl, 1 mM EDTA (pH 8.0), 0.25% deoxycholic acid) containing PhosSTOP phosphatase inhibitor and cOmplete proteinase inhibitor (both Roche, Basel, Switzerland). 20 µg of protein were run on a SDS-PAGE and semi-dry blotted onto a polyvinylidene difluoride membrane with the Trans-Blot Turbo System (all Bio-RAD, Hercules, CA). Membranes were blocked with 3% BSA in TRIS-buffered saline-tween buffer (TBS-T) and phospho-STAT6 (Y641), STAT6 (Rabbit Ab) or beta-actin (13E5) were applied as primary antibodies (all Cell Signaling Technology, Danvers, MA) at 4˚C overnight. Bound primary antibodies were detected by a HRP-linked polyclonal goat-anti-rabbit antibody (Rockland Immunochemicals, Limerick, PA) and the SignalFire Plus ECL reagent (Cell Signaling Technology) using the ChemiDoc imager (Bio-RAD). Membranes were stripped with the Restore PLUS Western Blot Stripping Buffer (Thermo Fisher Scientific) before a further primary antibody was applied.

## qRT-PCR

For qRT-PCR from stimulated cell cultures, BMDMs were lysed in RLT buffer (Qiagen, Hilden, Germany) with 0.1 M DTT and frozen at -80˚C overnight. RNA was isolated with the Quiagen RNeasy Kit (Qiagen) and cDNA was synthesized with the Superscript III reverse Transcriptase Kit (Thermo Fisher Scientific), both according to the manufacturer's protocol. For qRT-PCR from whole lung and whole small intestinal tissue, samples were collected in TRIsure (Bioline, Luckenwalde, Germany), shredded using 3 mm Tungsten Carbide Beads (Qiagen) in a Bead Ruptor (Omni International, Kennesaw, GA) with 3 cycles at 3.1 m/s for 15 sec and frozen at -80˚C overnight. RNA isolation was performed using the Chloroform/Isopropanol method and cDNA was synthesized with the Applied Biosystems High-Capacity cDNA Reverse Transcription Kit (Thermo Fisher Scientific). The SYBR Select Master Mix (Thermo) was used to perform qRT-PCR and these were measured either on a CFX-Connect instrument (Bio-Rad Laboratories, Inc., Hercules, CA) or a Applied Biosystems ViiA 7 Real-Time PCR System (Thermo Fisher Scientific). *Hprt* was used as a housekeeping gene. Primer sequences can be found in S1 Table.

## Histology

Tissue sections from the small intestine were fixed overnight in 4% para-formaldehyde (PFA; Sigma-Aldrich)) at 4˚C, followed by 15% (w/v) sucrose and 30% (w/v) sucrose in PBS and

embedded in tissue-freezing medium for cryo-sectioning (Leica, Wetzlar, Germany). 5–7 μm sections were cut at a cryostat followed by acetone fixation. Frozen sections were shortly thawed at 42˚C, rehydrated in PBS and blocked in TNB buffer (Perkin Elmer, Waltham, MA) containing 40 μg/mL anti-CD16/CD32 antibody (2.4G2, BioXcell) and 2% serum of the host the secondary antibodies were raised in. Sections were stained for Relm-α (unconjugated, Rabbit-anti mouse; abcam, Cambridge, UK), CD68 (Alexa Fluor 647- or Alexa Flour 488-conjugated, clone FA-11; BioLegend) or Arg1 (V-20 form Santa Cruz or D4E3M from Cell Signaling), Relm-β (unconjugated, Goat-anti-Mouse, R&D, Minneapolis, MN) Lysozyme (unconjugated Rabbit-anti-Mouse from Invitrogen), DCLK1 (unconjugated, Rabbit-anti-Mouse, abcam), Ki67 (D3B5 from Cell Signaling) and Epcam (Alexa Fluor 647- or Alexa Flour 488-conjugated, G8.8; BioLegend). Donkey-anti-Rabbit IgG coupled to Rhodamin-Red-X, Alexa Fluor 488 or Alexa Fluor 647 and Donkey-anti-Goat IgG coupled to Alexa Flour 488 (all Jackson ImmunoResearch Laboratories, West Grove, PA) were used as secondary antibody and cell nuclei were stained with 4,6 diamidino-2-phenylindole (DAPI) contained in Fluoroshield with DAPI histology mounting medium (Sigma-Aldrich). Stained sections were analysed on a confocal LSM700 microscope or a Axio VertA.1 microscope using the ZEN software (all Carl Zeiss, Jena, Germany).

## Flow cytometry

Flow cytometric analysis was performed in accordance to published guidelines [46]. PEC cells were collected in PBS. Whole blood was collected in FACS blood buffer (2% FCS (v/v), 0.1% NaN3 (v/v), 2000 U/L Heparin, in PBS) and applied to red blood cell lysis. Mesenteric lymph nodes were collected in PBS and gently meshed through a 70 μm filter to obtain single cell suspensions. For flow cytometry analysis of GC B cells, acid treatment was performed to remove cytophilic immunoglobulins from the cell surface. In brief, pellets of single cell suspensions were incubated with 1 mL acid buffer (0.085 M NaCl, 0.005 M KCl, 0.01 M EDTA, 0.05 M Na-Acetate, pH 4.0) for 2 min, followed by 0.5 mL FCS for 2 min and then washed twice with FACS buffer before surface staining. Unspecific staining was blocked by 5 min incubation with anti-CD16/CD32 antibody (2.4G2, BioXcell) prior to surface staining at 4˚C for 25 min. Fixable Viability Dye eFluor 506 (eBioscience, Thermo Fisher Scientific) was used to discriminate between dead and living cells. The following antibodies were used for surface stain: from eBioscience: CD4-FITC (RM4.4 and RM4-5), CD8-PE (53–6.7), CD11b-eFluor780 (M1/70), CD19-PE-Cy7 (1D3), F4/80-PerCP-Cy5-5 (BM8) and from BD (BD Bioscience, San Jose, CA): B220-BV711 (RA3-6B2), CD8-BUV737 (53–6.7), IgG1-BUV737 (A85-1), IgG2a/b-BUV395 (R2-40), Ly6G-BUV395 (1A8), PD-L2-APC (Ty25), Siglec-F-BV421 (E50-2440) and from BioLegend (SanDiego, CA): CD11b-FITC (M1/70), CD38-PerCP-Cy5-5 (90), F4/80-PE-Cy7 (BM8), GL7-PE (GL7). To determine the purity of isolated B cells and BMDM the following antibodies were used: F4/80-FITC or F4/80-BV421 (BM8), B220-PerCP-Cy5-5 or B220-BV421 (RA3-6B2) (all BioLegend) and CD19-eFluor780 (1D3, eBioscience). For IgG1-BUV737 (A85-1) intracellular staining was performed in addition. Cells were fixed and permeabilized with the BD Phosflow Fix and BD Phosflow Perm/Wash Buffer according to the company's protocol and stained intracellular for 25 min. Cells were acquired on a BD LSRFortessa (BD Bioscience).

## Quantification of histology

Briefly, the number of AAMs or of secretory epithelial subsets in histological sections was approximated by using the "AND" function in Fiji software. For Relm-α and Arg1 stainings, overlap of DAPI and CD68 defined macrophages and DAPI$^+$CD68$^+$ macrophages that overlap

with Relm-α or Arg1 were quantified. For Lysozyme and DCLK1, co-staining of these markers with DAPI was quantified. For quantification of Relm-β, DAPI⁺Epcam⁺ cells were defined as IECs and co-staining of these with Relm-β was quantified in Fiji (for S7 Fig). Due to technical difficulties, quantification of Relm-β for Fig 5 and S5 Fig, Relm-β⁺ spots in the epithelial region were counted manually here. Images with a magnification of 10x or 20x were used for Relm-α, Arg1 and Ki67. For Relm-β, DCLK1 and Lysozyme stainings a magnification of 5x was used.

## Statistical analysis

SigmaPlot Software (12.3, Systat Software, San Jose, CA) was used to analyse data for statistical significance and to create graphs. Data are displayed as mean + SEM. Statistically significant differences between two groups were determined by Student's t-test. For comparison of more than two groups, One-Way ANOVA or Two-Way ANOVA with Holm-Sidak *post-hoc* testing was used, depending on the number of factors included in the experimental setup, as indicated in the figure legends. In case normality or equal variance were not given, One-Way ANOVA was replaced by Kruskal Wallis test with Dunn's *post-hoc* test. The levels of significance used, were * $p \leq 0.05$, ** $p \leq 0.01$ and *** $p \leq 0.001$.

## Supporting information

**S1 Table. Primer sequences used for qRT-PCR analyses.**
(PDF)

**S1 Fig. Graphical abstract.**
(PDF)

**S2 Fig. (related to Fig 1).** AAMs in IL-4c-treated Mac-STAT6 mice.
(PDF)

**S3 Fig. (related to Fig 2).** *Nippostrongylus brasiliensis* (*Nb*) **infected Mac-STAT6 mice.**
(PDF)

**S4 Fig. (related to Fig 2).** Antibody and germinal center response of *Hpb* infected Mac-STAT6 mice.
(PDF)

**S5 Fig. (related to Fig 2).** Gating strategy for flow cytometry analysis.
(PDF)

**S6 Fig. (related to Fig 4).** Immune cell compartments in PEC and mLN of VillinCre_STAT6vt mice.
(PDF)

**S7 Fig. (related to Fig 5).** Immune cell compartments in PEC and mLN of VillinCre_IL-4Rα^fl/fl mice.
(PDF)

**S8 Fig. (related to Fig 6).** Epithelial cell histology for 4-13Tko mice.
(PDF)

**S1 Raw Data. Shows the values used to generate bar graphs.**
(XLSX)

## Acknowledgments

We thank Daniela Döhler for technical support, Natalie Thuma and Paul Haase for experimental advice, Ulrike Schleicher for providing Tie2Cre_Arg1$^{fl/fl}$ mice, Frank Brombacher for providing IL-4Rα$^{fl/fl}$ mice on B6 background, and members of the Voehringer lab for helpful discussions.

## Author Contributions

**Conceptualization:** Stefanie Westermann, Christoph Schubart, David Voehringer.

**Formal analysis:** Stefanie Westermann, Axel Dietschmann.

**Funding acquisition:** David Voehringer.

**Investigation:** Stefanie Westermann, Christoph Schubart, Kirstin Castiglione, Daniel Radtke.

**Project administration:** David Voehringer.

**Visualization:** Stefanie Westermann, Christoph Schubart.

**Writing – original draft:** Stefanie Westermann, Daniel Radtke, David Voehringer.

**Writing – review & editing:** Stefanie Westermann, Christoph Schubart, Axel Dietschmann, Daniel Radtke, David Voehringer.

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
