## [Decision Letter · Decision Letter 0]

28 Oct 2022

Dear Dr. Voehringer,

Thank you very much for submitting your manuscript "Th2-dependent STAT6-regulated genes in intestinal epithelial cells mediate larval trapping during secondary Heligmosomoides polygyrus bakeri infection" for consideration at PLOS Pathogens. As with all papers reviewed by the journal, your manuscript was reviewed by members of the editorial board and by several independent reviewers. In light of the reviews (below this email), we would like to invite the resubmission of a significantly-revised version that takes into account the reviewers' comments.

All reviewers agree that the topic of investigating epithelial cell mechanisms in intestinal helminth infections is interesting, but provide valid comments that should be addressed, These include better clarification in the manuscript text, addition of controls, and additional immune and parasite data.

We cannot make any decision about publication until we have seen the revised manuscript and your response to the reviewers' comments. Your revised manuscript is also likely to be sent to reviewers for further evaluation.

Sincerely,

Meera Goh Nair

Associate Editor

PLOS Pathogens

P'ng Loke

Section Editor

PLOS Pathogens

Kasturi Haldar

Editor-in-Chief

PLOS Pathogens

orcid.org/0000-0001-5065-158X

Michael Malim

Editor-in-Chief

PLOS Pathogens

orcid.org/0000-0002-7699-2064

All reviewers agree that the topic of investigating epithelial cell mechanisms in intestinal helminth infections is interesting, but provide valid comments that should be addressed, These include better clarification in the manuscript text, addition of controls, and additional immune and parasite data.

Reviewer's Responses to Questions

**Part I - Summary**

Reviewer #1: The article by Westermann et al describes the involvement of intestinal epithelial cells activation by IL-4/IL/13 in Heligmosmoides polygyrus trapping that occurs during secondary infection. In particular, they prove that macrophages arginase is not required for parasite trapping. This subject is of interest to the community, however I am not sure that the absence of macrophage requirement is proven nor does epithelial involvement in secondary protection is actually proven in the current version of the paper.

Number of repeats are sometimes too low for statistical analysis (example Fig3A) as 2 is not representative of a population.

Many control are missing :For figure 1, WT+IL-4c should be presented as well for worm count and for histology. For figure 4, primary control would be useful, as their is already protection happening in those mice during primary infection (both for worm count and histology).

Reviewer #2: The manuscript explores further how infection with Hpb is controlled by recall immunity. Work from the Gause group has previously demonstrated an important role for AAM in immune memory against Hpb.

In this study the authors explore this relationship further. Using mice expressing only STAT6 in macrophages they identify that when type 2 sufficiency is only present in Macs recall immunity is impaired. Moreover in an Arg1 -/- setting recall immunity is maintained. This is contrary to previous studies using arignase inhibitors. However, this may be explained by maintenance of increased expression of other AAM associated markers.

The authors then address the role of epithelieal cells in contributing to the effector response that promotes recall immunity. The authors identify that an effective epithelial memory response can be launched where STAT-6 is inherently expressed in epithelial cells and CD4 T cells are depleted.

The authors then show that in the absence of IL-4 and IL-13 from T cells recall immunity is impaired.

Therefore recall immunity is driven by a epithelial response which requires CD4 T cell promotion.

Reviewer #3: The authors use a well-characterized model of helminth infection in mice, with H. polygyrus, to establish requirements for larval trapping in the submucosa following secondary infection, which has been reported to require alternatively-activated macrophages (AAMs) and Arginase-1 (Arg-1), signaling through IL4/13 and STAT6. Using transgenic mouse constructs they report that Arg1 is in their hands not required, and that STAT6 activation in AAMs not sufficient for trapping. Rather, STAT6 activation in intestinal epithelial cells is required.

The data are somewhat one-dimensional, for example evaluating helminth immunity solely by granulomas with and without trapped larvae (no adult worm recovery or egg counts), presenting immunohistological snapshots without quantification or image analysis. Overall it offers an incremental contribution rather than a conceptual advance.

While this report shifts the focus towards an epithelial cell mechanism, there is no attempt to provide accompanying data on immune cell phenotypes in the different settings, and the paper does not advance far at any molecular level other than measuring the “usual suspects”. In particular, the authors miss the opportunity of using these mouse constructs (particularly the VillinCre-STAT6vt) for gene expression profiling and identification of the pathway within the epithelial population associated with immunity.

**Part II – Major Issues: Key Experiments Required for Acceptance**

Reviewer #1: - Antibodies have previously been shown to be important for trapping of Heligmosmoides (Hp) during secondary infection (10.4049/jimmunol.1401645). The author supplement their Mac-STAT6 with IL-4c to activate macrophage into "M2" however, this is known to not be sufficient for trapping of Hp larvae. Have the author verified if the antibody response against Hp is normal in their Mac-STAT6 mice, as a defect in antibody response would certainly explain why the mice fail to trap the parasite regardless of the M2 polarisation. Of note, to our knowledge IL-4c supplementation in a naive mice has not been shown to be sufficient for Hp trapping (the protocol the authors refer to is for macrophage activation in the lungs). This control naive mice + IL-4c+ Hp is required both for worm count and for histology before being able to conclude that STAT6 activation of macrophage is not sufficient for parasite trapping.

In FigS1, the authors actually report only 18.6% of peritoneal macrophage activated after IL-4c, which is really low. Only PDL-2 is used as activation marker (and not actually used anywhere else in the article), how are the classical M2 markers behaving :ARg1, RELMa, STAT6P ... As the IHC pictures are not presented per channel but only as merged, it is difficult to conclude if the macrophage are even activated in the STAT6MAc+IL4c.

- Figure 3, a quantification of macrophage number in the mucosa, based on histology or flow cytometry to validate that ARg1 depletion is not affecting macrophage recruitment is necessary to reach conclusion. One could quantify peritoneal macrophages as a surrogate.

- The use of the Villin_creSTAT6vt for secondary infection is very confusing. Indeed, the authors have previously shown that trapping takes place in those mice during primary infection and to a similar extent to what is reported here with the isotype control (about 60 granulomas with larvae). There seems to be no further protection due to reinfection. As such I fail to see the meaning of depleting CD4 T cells (previously shown to be required for secondary protection but not primary). Furthermore, the depletion of CD4Tcells in the control mice do not significantly impact the number of granulomas containing larvae. The conclusion of the authors "223 T cells are essential for larval trapping and activation of IECs in secondary Hpb infection." is not currently substantiated by their results.

The authors further state "However, the need for CD4+ 224 T cells could be overcome by

225 expression of a constitutively active form of STAT6 in IECs, emphasizing the prominent role

226 of STAT6-regulated genes in IECs in contributing to larval trapping", however they do not analyse what is the activation of those IECs, or their number. IHC of at least RELmb and an epithelial marker should be presented.

In absence of T cells, the author reports stronger RELmb expression, which contraditcs their claim that T cells are required for activation of IECs.

- Finally, the authors show that IL-4/IL-13 derived from T cells is required for protection. While this is well clear, they also state that this is required for IEC activation. However, here they only report total gene expression of RELMb in the intestine, which is not satisfactory to fully describe an epithelial activation. Could the authors look at proliferation of epithelial cells surrounding the larvae in the granuloma, show that those cells upregulate RELMb maybe even Phospholipase A2, which has already been shown to be important for protection during reinfection. RELMb role in primary expulsion has been established but not in trapping of the larvae during secondary. The authors could prove that RELMbKO mice are not protected during reinfection.

Reviewer #2: The paper is really interesting but I feel could be hard for researchers to grasp.

The authors should include a graphical illustration which highlights the findings from this study in the context of input to recall immunity to Hpb.

This should also highlight the authors thoughts on the importance of AAM. This is the potentially contentious issue with this manuscript. The authors do explain and justify extremely well the reasons for this difference. However, use of an illustration to highlight how this study introduces new insights into the role of epithelial cells and macs in recall immunity really will promote the message of this study.

Reviewer #3: Major Issues

1. The authors do not present any further information on the cellular responses in the different mouse models, which would be important data to understand the mechanisms at play. For example are all other compartments (CD4, ILC2, eosinophils…) normal in the mice unable to trap larvae? What is different about the cellular response in the VillinCre-STAT6vt mice?

2. The authors entitle Figure 2 “AAMs are not sufficient for larval trapping”; however the authors do not show data to document that in vivo AAM development is equal in the WT and Mac-STAT6 mice, they rely on in vitro phenotypes as shown in Figure 1.

3. Interpretation of the immunohistology sections relies on subjective judgement, which is questionable. For example, (lines 144-146) “Relm-a was abundantly expressed in IL4c-treated Mac-STAT6 mice” whereas comparison with the WT panel shows a very substantial reduction; hence it would appear (in contrast to the statement on lines 150-151) that both Relm-a and Arg-1 and Relm-a are compromised in the Mac-STAT6 mice. For objectivity, and also to ensure that the images are representative, quantification of large number of sections should be performed.

4. In Figure 3 the authors use a genetic KO of arginase-1 to follow up on published work with the pharmacological inhibitor BEC (Anthony et al, 2006 Ref 7). They find however that these mice retain the ability to trap larvae, in contrast to the earlier study. It is unfortunate that the authors did not also include WT mice treated with BEC to establish if the different result is due to the model, or other extraneous factors such as the animal facility being used. (Again Figure 3C is not accompanied by any quantitative analysis)

5. Generally, IL4/13 and STAT6 signaling alter the development of intestinal stem cells towards different specialized cell types. Hence an important dimension missing from Figure 4 is any analysis of goblet, Paneth and tuft cells in the different experimental groups.

6. For Figure 5, the authors use IL4/13 KO mice, both as global and T cell-specific Kos to measure the impact on helminth immunity; as would be predicted, immunity is ablated in both. The insight offered by this new experiment is very limited, as in the absence of T cell cytokines from the outset of infection it is only to be expected that immunity will fail to develop ; it would be more useful to ascertain whether T cell cytokines are required at the time of challenge infection.

**Part III – Minor Issues: Editorial and Data Presentation Modifications**

Reviewer #1: - line 104 : the authors state : "Contrary to expectations and for reasons that remain unclear, STAT6 in the BMDMs of Mac-STAT6 mice was not constitutively phosphorylated". It is unclear to me while constitutive activation of STAT6P would have been expected, as this is usually regarded as M2 polarisation marker.

- For all histology, could the author separate the channels, as it would help see colocalization.

- In figure 2C both in WT and in STAT6Mac there is a strong RELMa staining that do not colocolize with CD68, potentially of epithelial origin. However the authors then use RELMa as a marker of macrophage only as opposed to RELmb as a marker of IEC. Staining for EpCAM or related IEC marker might be helpful in the current setting.

- Fig2D:Why is there Arg1 expression in STAT6KO mice ? Does this actually colocolize with a macrophage marker ? In IL4RKO mice, arginase stain has been previously reported to not be present (10.1371/journal.ppat.1003771). There also seems that there is less Arg1 around the larvae in the MACSTAT6+ IL4c that in the STAT6KO which is confusing. A quantification of the staining would be helping with interpretation. Outlining the larvae as well (as they are quite autofluorescent) would help as well.

-There have been some recent papers with improve cell isolation protocol during Hp infection ( Ferrer-Fon et al, 2020 elife; Bouchery et al, 2017, current protocol). Flowing the intestinal macrophage might thus give a better idea of their actual polarisation.

- Besides, AAMs, defined here by Relm-α and

180 CD68 co-expression, are surrounding the granuloma in a similar amount in mice of both

181 genotypes (Fig 3C, left panel.

Please quantify based on images.

Reviewer #2: (No Response)

Reviewer #3: Minor Issues

1. Line 63 Reference 8 describes work with N brasiliensis not H polygyrus as stated here.

2. Lines 65-67 (and throughout the manuscript) – there is no mention of tuft cells

3. Lines 69-78 the text describing previous work lacks insight or detail and any key differences between the studies are not defined. At least 2 references are missing:

G. Chen, S. H. Wang, J. C. Jang, J. I. Odegaard and M. G. Nair: Comparison of RELMα and RELMβ Single- and Double-Gene-Deficient Mice Reveals that RELMα Expression Dictates Inflammation and Worm Expulsion in Hookworm Infection. Infect Immun, 84(4), 1100-1111 (2016) doi:10.1128/IAI.01479-15

H. M. Batugedara, J. Li, G. Chen, D. Lu, J. J. Patel, J. C. Jang, K. C. Radecki, A. C. Burr, D. D. Lo, A. R. Dillman and M. G. Nair: Hematopoietic cell-derived RELMα regulates hookworm immunity through effects on macrophages. J Leukoc Biol (2018) doi:10.1002/JLB.4A0917-369RR

4. Line 79 diverse not divers

5. Figure 1 B : why do most graphs show large effects but no significance?

6. Line 335 where not were

7. Lines 395-403 Methods detailing B cell purification do not appear to be necessary as no data are shown.

PLOS authors have the option to publish the peer review history of their article (what does this mean?). If published, this will include your full peer review and any attached files.

Reviewer #1: No

Reviewer #2: No

Reviewer #3: No
---

## [Decision Letter · Decision Letter 1]

13 Mar 2023

Dear Dr. Voehringer,

We are pleased to inform you that your manuscript 'Th2-dependent STAT6-regulated genes in intestinal epithelial cells mediate larval trapping during secondary Heligmosomoides polygyrus bakeri infection' has been provisionally accepted for publication in PLOS Pathogens.

Best regards,

Meera Goh Nair

Academic Editor

PLOS Pathogens

P'ng Loke

Section Editor

PLOS Pathogens

Kasturi Haldar

Editor-in-Chief

PLOS Pathogens

orcid.org/0000-0001-5065-158X

Michael Malim

Editor-in-Chief

PLOS Pathogens

orcid.org/0000-0002-7699-2064

Both Editor and Reviewer agree that the resubmitted manuscript is improved and will be of interest to the field and journal readership.

Reviewer Comments (if any, and for reference):

Reviewer's Responses to Questions

**Part I - Summary**

Reviewer #1: The manuscript by Westermann et al, has substantially improved with revision. It now elegantly show that IECs are involved in protection against Hp during secondary infection. Arginase-1 depletion with a deficient model was shown not to be required for protection, and AAM polarization in absence of STAT6 signaling in IECs is not sufficient to perform killing. Further deciphering of the interplay between epithelial cells and macrophages will be extremely interesting to understand how killing is mediated.

**Part II – Major Issues: Key Experiments Required for Acceptance**

Reviewer #1: I am now satisfied with the answer to comments.

Of note, for figure 3A, 2 mice per genotype is still low for any type of interpretation regarding of the change the authors have made to the graph ( a minimum of 3 individuals is always required to perform statistical analysis and to sample a population, and this is a mere minimum)

**Part III – Minor Issues: Editorial and Data Presentation Modifications**

Reviewer #1: no more issues to report

PLOS authors have the option to publish the peer review history of their article (what does this mean?). If published, this will include your full peer review and any attached files.

Reviewer #1: No

---

## [Editor Report · Acceptance letter]

30 Mar 2023

Dear Dr. Voehringer,

We are delighted to inform you that your manuscript, "Th2-dependent STAT6-regulated genes in intestinal epithelial cells mediate larval trapping during secondary Heligmosomoides polygyrus bakeri infection," has been formally accepted for publication in PLOS Pathogens.

Best regards,

Kasturi Haldar

Editor-in-Chief

PLOS Pathogens

orcid.org/0000-0001-5065-158X

Michael Malim

Editor-in-Chief

PLOS Pathogens

orcid.org/0000-0002-7699-2064